# Dissipation of Emamectin Benzoate Residues in Rice and Rice-Growing Environments

**DOI:** 10.3390/molecules25030483

**Published:** 2020-01-23

**Authors:** Ligang Deng, Lu Chen, Shuai Guan, Junhua Liu, Jingyun Liang, Xia Li, Zengmei Li

**Affiliations:** Key Laboratory of Test Technology on Food Quality and Safety of Shandong Province, Institute of Agricultural Quality Standards and Testing Technology Research, Shandong Academy of Agricultural Sciences, 202 Gongyebeilu Road, Jinan 250100, China; zhbsdengligang@shandong.cn (L.D.); zhbschenlu@shandong.cn (L.C.); zhbsguanshuai@shandong.cn (S.G.); zhbsliujunhua@shandong.cn (J.L.); zhbsLiangjingyun@shandong.cn (J.L.)

**Keywords:** emamectin benzoate, rice, residue, degradation, UPLC-MS/MS

## Abstract

The experiment developed the ultra-high-performance liquid chromatography–tandem mass spectrometry (UPLC/MS/MS) method for testing emamectin benzoate, and studied the metabolism of emamectin benzoate in rice plants and rice-growing environments via application of this testing method. The dissipation curve of emamectin benzoate standard substance was good at 0.5–200 μg L^−1^, and its correlation coefficient was greater than 0.99. In the concentration range of 0.1–50 μg kg^−1^, the average recovery rate of plants, soil, and field water was 82 %–102 %, and relative standard deviation (RSD) was between 0.3 % and 15.9 %. Half-lives in rice plants and soil were 0.8–2.8 days and 1.9–3.8 days, respectively, and emamectin benzoate was not detected in rice or rice hull. The experiment showed that emamectin benzoate is harmless to human health at the concentration recommended by the manufacturer.

## 1. Introduction

Emamectin benzoate is a kind of macrolide compound made by modifying the chemical structure of the raw material, avermectin B1, and is a novel and efficient semi-synthetic pesticide [1]. It is neurotoxic, and has a significant control effect on Lepidoptera, Homoptera, Coleoptera, and other pests [2,3]. Due to its highly efficient broad spectrum and long period of validity, it is not easy for treated pests to develop resistance, and it is widely used in the pest control of fruits, vegetables, and field crops as an alternative to highly toxic pesticides.

The careless use of emamectin benzoate could cause environmental pollution and serious harm to other, non-target organisms. Research has shown that if rats are exposed to high doses of emamectin benzoate during pregnancy and lactation, the offspring display significant neurotoxicity, which indicates that the neurotoxic effect of emamectin benzoate is particularly prominent in animals [4]. In addition, other reports have shown that the LD_50_ of emamectin benzoate to bees, quail, Bombyx mori, and zebrafish is very low [5,6,7], which indicates that emamectin benzoate is highly toxic or particularly highly toxic to these organisms. With this in mind, maximum residue limits (MRLs) of emamectin benzoate on the crops have been defined in many countries. The MRLs in cabbage and kale are 0.05 mg kg^−1^ and 0.1 mg kg^−1^ in China; the MRL in rice is 0.01 mg kg^−1^ in the EU and 0.02 mg kg^−1^ in Brassica vegetables in Australia.

Multiple methods of analyzing emamectin benzoate have been reported. High-performance liquid chromatography separation (HPLC) has been used in conjunction with a UV detector (UVD) [8,9,10] and a fluorescence detector (FLD) [11,12,13,14,15]. However, the sensitivity of the UVD was low and failed to meet the requirements of the standard, and the derivatization condition of the fluorescence detection method was unstable, which was unconducive to the analysis of a batch sample. In recent years, UPLC-MS/MS has become a rapid and frequently used determination method for an analysis of avermectins [16,17]. Krogh [18] reported the detection of avermectin drugs in water, sediment, and soil using HPLC-MS/MS. In this research, a simple, relatively fast, and efficient UPLC-MS/MS method for the residue analysis of emamectin benzoate was developed. This study optimized a sample pretreatment method suitable for the analysis of rice, rice stems, water, and even soil, and studied the residual changes and degradation of emamectin benzoate in rice-growing environments.

## 2. Results and Discussion

### 2.1. Optimization of the Liquid Chromatogram–Mass Spectrum Condition

In this experiment, a C_18_ chromatographic column was used for separation. Ammonium acetate was added into the mobile phase to promote the ionization of the compound, which significantly improved the response value of the mass spectrum. The relative molecular weight of the emamectin benzoate was 1008.24, from which the emamectin (C_49_H_75_NO_13_, molecular weight of 886.4) and the benzoate (C_7_H_6_O_2_, molecular weight of 122) can be completely dissociated in water, so this experiment regarded emamectin as the detection object. The mass spectrum level obtained by the parent ion scan (MS scan) showed that emamectin existed in the form of a molecular ion peak [M-C_7_H_5_O_2_-H]^+^ (*m/z* 886.4), and the capillary voltage, desolvation temperature, and other parameters were optimized to reach the highest intensity of the parent ions. The secondary mass spectrum was obtained through a product ion scan (daughter scan) (Figure 1). In Figure 1, the main characteristic fragment ions were *m/z* 158.35 and *m/z* 82.74 after splitting, presumably corresponding to [C_7_O_2_H_11_-(CH_3_)-NH_2_]^+^ and [C_6_H_10_]^+^ peaks. The parameters of the cone voltage and collision energy and other conditions were further optimized in order to obtain the optimum instrument parameters for the maximum sensitivity to the characteristic fragment ions.

### 2.2. Linear Equation and Matrix Effects

Matrix interference, including ion suppression and ion enhancement, is an important consideration for UPLC-MS/MS, and can influence the accuracy, sensitivity, linearity, and stability of the quantification [19,20,21]. In order to ascertain the matrix effect of the sample, the standard solution curve and matrix-matched standard were compared at six concentrations levels: 0.5, 5, 10, 50, 100, and 200 μg L^−1^. Calibration curves were constructed by plotting the integrated peak areas (Y) against the concentrations of compound (X). The results of the linear regression and regression equation are shown in Table 1.

The slope ratio between the slope of the matrix-matched standard curves and the slope of the standard solution curves reflects the effect of matrix interference [22]. The results show that the ion in the soil samples was strongly suppressed, which may have been due to the large amount of inorganic salts, organic matter, microorganisms, and other components in the soil, which competed for ionization with the target analytes to produce a strong matrix inhibition effect, while the result measured by the standard curve using pure solvent was much lower than the actual result. The rice stem samples also showed strong matrix inhibition effects. These results were consistent with research about the matrix effects of abamectin in paddy rice [16]. The field water samples showed slight matrix enhancement effects, while the result measured by the standard curve using pure solvent was higher than the actual result. Therefore, in the process of actual sample detection, the standard curves of the matrix solution should be used for a quantitative analysis to avoid a matrix effect and ensure a more accurate quantitative analysis.

### 2.3. Sensitivity, Accuracy, and Precision of Method

The recovery results for blank rice stems, soil, and field water samples at three concentration levels of 0.1, 10.0, and 50.0 μg kg^−1^ are shown in Table 2. The average recovery ranged from 82% to 102%, in line with the accuracy requirement of the analysis, and the relative standard deviation (RSD) was between 0.3% and 15.9%. The minimum limit of the quantitation (LOQ) of emamectin benzoate was 0.1 μg kg^−1^ (signal-to-noise ratio ≥ 10), which was fully able to meet the test requirements.

### 2.4. Metabolism of Emamectin Benzoate in Rice Environment

Table 3 indicates the degradation of emamectin benzoate in the rice stems and soil.The degradation law was in line with the exponential regression equation C = C_o_e^−kt^, while the degradation curve is shown in Figure 2. The original residue of emamectin benzoate in the rice stems was 35.0–64.2 μg kg^−1^. As expected, a gradual and continuous degradation was observed after application. The metabolic half-life of emamectin benzoate was 0.8–2.8 d in rice stems, which classifies it as a readily biodegradable pesticide. The main deterioration was observed to take place within the first week, as can be seen in Figure 2a. The results of this study are consistent with several previous studies, in which the half-life of emamectin benzoate in cucumber and soil was found to be 0.1–1.5 d [23], while the half-life of emamectin benzoate in vegetables and cabbage was found to be 1.10–2.10 d [24].

The paddy environment is a complex system, and the residue of emamectin benzoate in the field water and soil showed subtle changes after the treatment spraying of the rice stems. In the field water, the original amount of emamectin benzoate deposited was 0.21–1.40 μg kg^−1^ after 1 h of application and the maximum residual amount was 0.40 μg kg^−1^ after 6 h, while the residual amount was undetectable after 1 day. However, in the soil, the amount of emamectin benzoate deposited was not detected after 1 h of spraying, and it gradually increased to reach 3.64–7.40 μg kg^−1^ in 1–2 days. The residual emamectin benzoate in the soil was then gradually degraded, and the half-lives in soil were 1.9–3.8 d. Figure 2 shows the residue changes in the soil. Emamectin benzoate decomposes into emamectin in water. It is difficult to detect emamectin in a paddy water environment with a large volume of water and relatively low concentration. Emamectin easily adsorbs onto soil, and gradually migrated from the water into the soil; therefore, the residue of emamectin in the soil gradually increased and decreased in the water. Sun et al. [12] reported that the migration rate of emamectin benzoate in different water–soil systems was paddy soil > red soil > yellow cinnamon soil > black soil, with the strongest migration in paddy soil.

### 2.5. Final Residue of Emamectin Benzoate in the Rice Environment

The results of the final residue tests are shown in Table 4. The residues of emamectin benzoate in the rice stems collected 7 days and 14 days from the last spraying were 0.1–0.97 μg kg^−1^, and the maximum residue amount in the soil samples was 3.17 μg kg^−1^, which was much lower than the MRL (China, GB2763), while the residues of emamectin benzoate in rice and rice husks were all undetectable (lower than LOQ). Based on the results of this test, emamectin benzoate is safe for rice when it is sprayed in the recommended dosage.

## 3. Materials and Methods

### 3.1. Materials and Standards

Acetonitrile and methanol were HPLC grade (Fisher Scientific, Pittsburgh, PA, USA). Ammonium acetate was HPLC grade, 99.6 % ammonium benzoate standard (Sigma-Aldrich Chemie GmbH, Saint Louis, MO, USA). Analytical pure ethyl acetate, phosphoric acid, and sodium chloride (Sinopharm Chemical Reagent Co., Ltd, Beijing, China.), and PLEXA PCX solid-phase extraction columns ( Varian Corporation, Palo Alto, CA, USA).

### 3.2. Site Description and Sample Collection

Field trials, including a dynamic test and final residue measurements, were carried out in Shandong, Henan, and Zhejiang provinces, China, in 2010. Three treatments were established for the field trial, each of which was replicated three times with an area of 30 m^2^, divided by an isolation strip in order to avoid cross-contamination. In the decomposition dynamic test, emamectin benzoate was sprayed once onto the surface of the rice stem and water/soil surface at a dosage of 22.5 g a.i.hm^2^ in the early period of rice heading, after which the rice stems, field water, and soil were collected at intervals of 1 h and 6 h and 1, 2, 3, 5, 7, 14, 21, and 30 days. The final residue field trial was carried out at two dosage levels: 15 g a.i. hm^2^ (recommended dosage) and 22.5 g a.i. hm^2^ (1.5 times the recommended dosage), which were sprayed once and twice, respectively. The rice stems, spikes of rice (brown rice and rice husk), and soil samples were gathered 7 days and 14 days from the last spraying. All the samples were stored at −20 °C until they were analyzed.

### 3.3. Sample Preparation

All the samples were prepared before the analysis. The rice stems were cut into small pieces of less than 1 cm. The brown rice and rice husk were picked out after the threshing of the rice spike samples, and ground to a coarse powder with a vegetation disintegrator. The soil samples were sifted through a 40 mesh sieve.

### 3.4. Extraction and Purification

Protocol was modified from that used in research by K.A. Krogh [15]. A 10 g sample was weighed into an 80 mL centrifuge tube and extracted with 1 mL acetonitrile and 15 mL of ethyl acetate. The supernatant was collected after 10,000 rpm high-speed homogenization for 1 min. Next, 1 g NaCl was added to the supernatant, and the ethyl acetate layer on the upper was collected from 4000 rpm for 5 min and then added to 15 mL ethyl acetate, and the remaining residue was re-extracted. The extract liquor was combined twice to purify it.

The PCX Solid Phase Extraction cartridge was first conditioned successively with 5 mL of 1% ammonium acetate methanol, 1% phosphoric acid methanol, water, methanol, and ethyl acetate, and then immediately added to the sample solution when the solution liquid level flowed to the adsorption surface of the filler. After all the solution had been transferred to the extraction column, the solid phase extraction column was eluted with 3 mL of ethyl acetate, and the eluent was discarded. Finally, it was eluted with 10 mL of 1% ammonium acetate–methanol solution and the eluent was collected in a round-bottomed flask.

The collected eluent was rotary evaporated to near dryness, then dissolved in 1 mL of methanol. Next, 4 mL of water and finally 5 mL of ethyl acetate were added to the extract. After the vortex oscillation, it was left to stand for 10 min before collecting the supernatant. The lower layer of the solution was then added to 5 mL of ethyl acetate to be extracted again, and the extracted liquor was combined twice and evaporated to dry with nitrogen, then dissolved in 2 mL of methanol. The extract was filtered through a 0.22 μm pore membrane filter and transferred into a 1.5 mL glass vial for analysis.

### 3.5. UPLC-MS/MS Analysis

A Quattro PremierTM triple quadrupole mass spectrometer (Micromass, Milford, MA, USA) was used for the MS/MS analysis in ESI (+), and the data were acquired in a multiple reaction monitoring (MRM) mode. The source working conditions were as follows: capillary voltage: 3.5 KV; photomultiplier voltage: 650 V; ion source temperature: 110 °C; desolvation temperature: 400 °C; cone gas and desolvation gas: nitrogen, 99.99% purity; flow rate: 600 L h^−1^; collision chamber vacuum degree: 7.22 × 10^−4^ (mbar); and mass analyzer vacuum degree 8.00 × 10^−6^ (mbar). Other conditions in MRM mode are listed in Table 5.

The spectrum was the total ion chromatogram (TIC) and the selective ion chromatogram of emamectin benzoate. The interscan delay time and the inter-channel delay time were set as 0.1 min and 0.02 s, respectively. The analyte was separated on a reverse-phase ACQUITY UPLC BEH C18 column (50 × 2.1 mm, id 1.7 μm) and eluted by mobile phases of A (acetonitrile) and B (10 mmol L^−1^ ammonium acetate). The elution program was a gradient elution: initial 10% A: 0–2 min; A linear increase to 98%: 2–4.0 min; A linear decrease to 10%: 4–4.5 min, and then a system balance. Data acquisition was carried out by MasslLynx V4.0 software. The UPLC-MS/MS chromatogram of the standard solution of emamectin under these conditions is shown in Figure 3.

### 3.6. Statistical Analysis

Data calculations for rice stems, water, and soil samples used each matrix standard curve for quantification, applying the external standard method within the linear range of the measured object. The dissipation process followed the first-order kinetic reaction. The degradation rate constant and half-lives were calculated using a first-order rate equation: C_t_ = C_0_e^−kt^, where C_t_ is the residue concentration of pesticide at the time of t, C_0_ is the initial concentration of pesticide after application, and k is the degradation rate constant (d). The half-life (t_1/2_) was calculated from the k value for each experiment (t_1/2_ = ln 2/k).

## 4. Conclusions

The experiment used the ultra-high-performance liquid chromatography–mass spectrometry method (UPLC-MS/MS) to study emamectin benzoate. The method was simple, reliable, and highly sensitive, and was applied in the present work to study the metabolism and degradation law of emamectin benzoate in the rice-growing environment. After spraying the pesticide, emamectin benzoate degraded fast in rice stems, and its half-life was 0.8–2.8 d. In the planting environment, it gradually migrated from the water into the soil, and the half-life in the soil was 1.9–3.8 days, which defines it as a readily biodegradable pesticide. The results showed that emamectin benzoate degraded fast in rice stems and the planting environment, including water and soil. The present study provides reliable techniques, tools, and references for a risk assessment of emamectin benzoate pesticide in a rice paddy environment. The final residue results showed that emamectin benzoate is harmless to human health at the concentration recommended by the manufacturer. This work will help to provide adequate monitoring of residues on a quantitative basis for the revision of the application of this pesticide to rice crops.

## Figures and Tables

**Figure 1 molecules-25-00483-f001:**
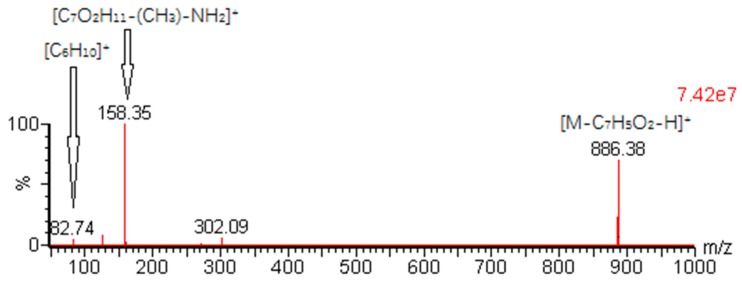
Mass spectrum of emamectin by product scan.

**Figure 2 molecules-25-00483-f002:**
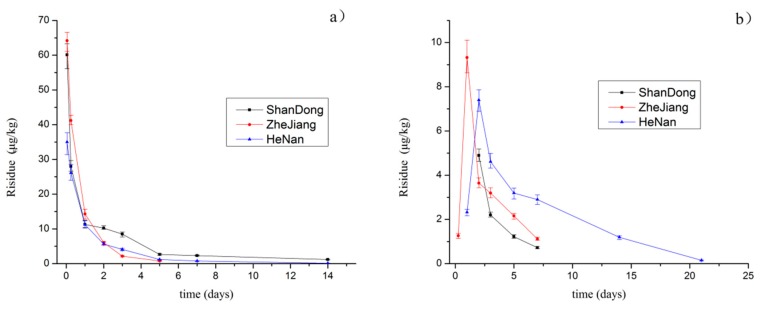
Dissipation curve of emamectin benzoate in rice stem (**a**) and soil (**b**).

**Figure 3 molecules-25-00483-f003:**
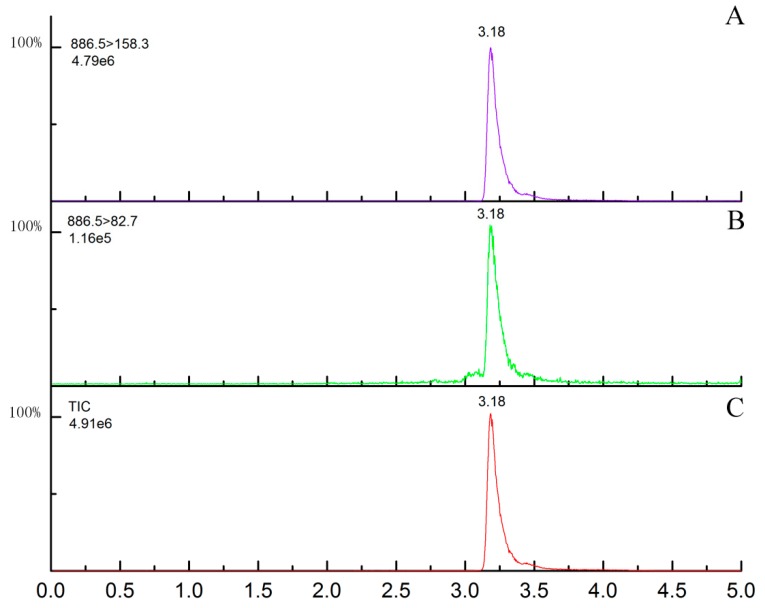
MRM (multiple reaction monitoring) mode chromatograms of emamectin standard (10 μg L^−1^). (**A**) The quantitative ion chromatogram; (**B**) the qualitative ion chromatogram; (**C**) the total ion chromatogram.

**Table 1 molecules-25-00483-t001:** Calibration curve of emamectin benzoate in rice stems, soil, and field water (*n* = 5).

	Calibration Curve	Determination Coefficients (R^2^)	Slope Ratio (Matrix/Methanol)
Methanol	Y = 77,120.9X + 364,327	1.0000	-
Rice stem	Y = 61,680.74X + 1517.49	0.9999	0.7998
Paddy soil	Y = 50,256.8X + 218.418	0.9998	0.6517
Paddy water	Y = 91,379.5X + 46,277.7	0.9999	1.1849

**Table 2 molecules-25-00483-t002:** Recovery and relative standard deviations (RSDs) (*n* = 5) of emamectin benzoate at the spiked level in blank sample.

Samples	Spiked Level (μg kg^−1^)	Average Recovery (%)	RSD (%)	LOQ (μg kg^−1^)
Rice stem	0.1	102	8.2	0.1
	10.0	97.4	2.2	0.1
	50.0	98.6	0.3	0.1
Paddy soil	0.1	94	5.8	0.1
	10.0	99.4	1.1	0.1
	50.0	96.6	1.2	0.1
Paddy water	0.1	82	15.9	0.1
	10.0	98.8	1.1	0.1
	50.0	99.6	0.6	0.1

**Table 3 molecules-25-00483-t003:** Dynamic degradation equation of emamectin benzoate in rice stem and soil (*n* = 5).

Test Sites	Rice Stem				Soil		pH
	Digestion Equation	Correlation Coefficient (r)	Half-Life (Days)	Digestion Equation	Correlation Coefficient (r)	Half-Life (Days)	
Shan Dong	C_T_ = 20.922e^−0.2494T^	−0.7779	2.8	C_T_ = 8.0994e^−0.3601T^	−0.9432	1.9	6.8
Zhe Jiang	C_T_ = 45.335e^−0.8901T^	−0.9618	0.8	C_T_ = 8.0103e^−0.2637T^	−0.9651	2.6	5.6
He Nan	C_T_ = 17.653e^−0.4002T^	−0.9310	1.7	C_T_ = 9.7098e^−0.1848T^	−0.9579	3.8	5.9

**Table 4 molecules-25-00483-t004:** The final residue of emamectin benzoate in plants, rice hull, rice, and soil.

Site	Dosage(g a.i./hm^2^)	Number of Times Sprayed	Residue (µg kg^−1^)							
			7 Day				14 Day			
			Rice Stem	Paddy Soil	Rice Husk	Rice	Rice Stem	Paddy Soil	Rice Husk	Rice
Shan Dong	15	1	0.97	0.25	<0.1	<0.1	0.50	<0.1	<0.1	<0.1
		2	0.43	0.57	<0.1	<0.1	0.27	<0.1	<0.1	<0.1
	22.5	1	0.23	0.63	<0.1	<0.1	0.20	<0.1	<0.1	<0.1
		2	0.37	3.17	<0.1	<0.1	0.43	<0.1	<0.1	<0.1
Zhe Jiang	15	1	0.13	<0.1	<0.1	<0.1	<0.1	<0.1	<0.1	<0.1
		2	<0.1	<0.1	<0.1	<0.1	<0.1	<0.1	<0.1	<0.1
	22.5	1	0.61	<0.1	<0.1	<0.1	0.1	<0.1	<0.1	<0.1
		2	0.37	<0.1	<0.1	<0.1	0.17	<0.1	<0.1	<0.1
He Nan	15	1	0.1	<0.1	<0.1	<0.1	<0.1	<0.1	<0.1	<0.1
		2	0.1	<0.1	<0.1	<0.1	<0.1	<0.1	<0.1	<0.1
	22.5	1	0.1	<0.1	<0.1	<0.1	<0.1	<0.1	<0.1	<0.1
		2	0.1	<0.1	<0.1	<0.1	<0.1	<0.1	<0.1	<0.1

**Table 5 molecules-25-00483-t005:** ESI^+^ MS/MS parameters for the determination of emamectin benzoate B1a.

Compound	Molecular Formula	Retention Time (min)	Parent Ion(*m/z*)	Daughter Ions(*m/z*)	Dwell Time (s)	Cone Voltage (V)	Collision Energy (eV)
Emamectin benzoate B1a	C_49_H_75_NO_13_·C_7_H_6_O_2_	3.18	886.4	158.3 *	0.1	50	15
				82.7		45	27

Note: Quantitative product ions. * The response value is more significant and can be used for quantitative detection

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
