# Peer review of "Dissipation of Emamectin Benzoate Residues in Rice and Rice-Growing Environments"

_molecules, 2020, doi:10.3390/molecules25030483_

Round 1

Reviewer 1 Report

The authors reported about a metabolic study of the pesticide, emamectin benzoate, in rice and rice cultivation field like paddy soil and water. The manuscript topic is suitable to “Molecules” and is also relatively well organized. But the manuscript needs to further improve a scientific quality about results section. Some comments and concerns are addressed the below, and hope there are helpful to improve the scientific quality of this manuscript.

Line 10: change to ultra, not super Line 46: what does it mean? Need to describe more detail about “efficient” how fast, sensitive, accurate, reproducible, and others. Line 58: must use small letter in the sentence, not capital letter. So change to “The level in mass….” Line 62: changed to product ion. Also, Fig.2 is not matched to this description. Need to correct. Figure 1: need to include fragmentation patterns for emamectin benzoate Figure 2: how many replicates? Need to add standard deviation bars at all data points. Also, add the results for water with adjusting scale bar of the X-axis like log scale. Line 74: change to this study…… Table 1: I think it is better to show as Figure with this information. please change to the Figures of calibration curves of external and matrix-matched. Table 2: how to determine LOQ? Nedd to describe it. Line 106: use small letter for “….week as can….” Table 3: mow many replicates? Also, need to describe the soil physicochemical features of 3 regions. Line 117: Fig. 3 is not for the results of soil. Need to correct. Line 128: describe the detail level of MRL (for China or others) Table 4: specify the unit for dosage (g.a.i.hm2)?? Line 134: describe all material sources as like city, state, country Line 142: samples were collected 2010 ??, when did authors analyze these samples? This is very important, so describe in detail. Line 157: add references. Line 163: why did authors use the PCX SPE catridge? is this eliminated (non-)polar interferences in the sample aliquot?? Need to specify. Table 5: adjust the first line in the Table 5. Figure 3. Need to describe in detail about Figure 3 in the manuscript. line 206: describe more detail how reliable, sensitive accurate? Discussion: I think it is also strongly need to discuss detection of other metabolites of emamectin benzoate. Language editing is required.

-The end-

Author Response

Point 1:Line 10: change to ultra, not super

Response 1:Line 10:The “super” has modified to “ultra”.

Point 2:Line 46: what does it mean? Need to describe more detail about “efficient” how fast, sensitive, accurate, reproducible, and others.

Response 2:Line 46: Ultra-high performance liquid chromatography-tandem mass spectrometry has a short analysis time and is completed in 5 minutes; The detection limit is low, the sensitivity is high, LOQ can reach 0.1μg/kg; accurate-recovery rate is good, average recovery rate is 82% -102%; reproducible-RSD is 0.3% -15.9% .

Point 3:Line 58: must use small letter in the sentence, not capital letter. So change to “The level in mass….”

Response 3:Line 58: English writing has been regulated.

Point 4:Line 62: changed to product ion. Also, Fig.2 is not matched to this description. Need to correct.

Response 4:Line 62: Daughter ion modified to product ion.  Fig. 2 in the manuscript has modified to Fig.1 .

Point 5: Figure 1: need to include fragmentation patterns for emamectin benzoate Figure 2: how many replicates? Need to add standard deviation bars at all data points. Also, add the results for water with adjusting scale bar of the X-axis like log scale.

Response 5: Figure 2 has been modified and redrawn. We have three replicates, but the experimental results show that the residual amount in water is less than the limit of quantification, so the digestion residual spectrum of water is not listed.

Point 6: Line 74: change to this study……

Response 6:Line 74: The paper has been modified to this study.

Point 7:Table 1: I think it is better to show as Figure with this information. please change to the Figures of calibration curves of external and matrix-matched.

Response 7:Standard curve table can express experimental data more intuitively and accurately.

Point8:Table 2: how to determine LOQ? Nedd to describe it.

Response 8:We have added explanation after LOQ (signal-to-noise ratio≥10).

Point 9:Line 106: use small letter for “….week as can….”

Response 9:Line 106: English writing has been regulated.

Point 10:Table 3: mow many replicates? Also, need to describe the soil physicochemical features of 3 regions.

Response 10:Table 3 has been refined.

Point 11:Line 117: Fig. 3 is not for the results of soil. Need to correct.

Response 11:Line 117:Fig. 3 in the manuscript has been modified to Fig.2 .

Point 12: Line 128: describe the detail level of MRL (for China or others)

Response 12:Line 128: The MRL was according to Chinese GB2763.

Point 13:Table 4: specify the unit for dosage (g.a.i.hm2)??

Response 13:Table 4 Unit format has been modified.

Point 14: Line 134: describe all material sources as like city, state, country

Response 14:Line 134: Materials and reagents have been written as required

Point 15:Line 142: samples were collected 2010 ??, when did authors analyze these samples? This is very important, so describe in detail.

Response 15:Line 142:We completed the experiment in 2011, but the manuscript was submitted recently.

Point 16:Line 157: add references.

Response 16:Line 157: we have added references.

Point 17: Line 163: why did authors use the PCX SPE catridge? is this eliminated (non-)polar interferences in the sample aliquot?? Need to specify.

Response 17:Line 163:Emamectin Benzoate has an amine group in the molecular structure, which can interact with the PCX column, and other impurities flow out of the PCX column to achieve selective separation. Tests show that the PCX column has good purification effect.

Point 18: Table 5: adjust the first line in the Table 5.

Response 18:Table 5 :Table font has been modified.

Point 19:Figure 3. Need to describe in detail about Figure 3 in the manuscript.

Response 19:Figure 3. We have added a description to Figure 3.

Point 20: line 206: describe more detail how reliable, sensitive accurate? Discussion: I think it is also strongly need to discuss detection of other metabolites of emamectin benzoate. Language editing is required.

Response 20: line 206: In 2010, China suggested that the residue of metavidin in crops be defined as B1a. In this experiment, the study of metabolites was not involved. The next step is to focus on the trend of metabolite residues.

Reviewer 2 Report

The paper is interesting for the publication in the journal only a little correction is need.

Abstract Line 10 write "Ultra-performance liquid chromatography tandem mass spectrometry (UPLC/MS/MS)" and not "super-high-performance liquid Chromatogr-amtandem massa spectrometry (UPLC/MS/MS)".

Introduction line 47 - UPLC/MS/MS not HPLC/MS/MS

Figure 3 - explain the figure what refers, for example with letter and explain the meaning of each chromatogram.

Author Response

Point 1:Abstract Line 10 write "Ultra-performance liquid chromatography tandem mass spectrometry (UPLC/MS/MS)" and not "super-high-performance liquid Chromatogr-amtandem massa spectrometry (UPLC/MS/MS)".

Response 1: Abstract Line 10 The "super-high-performance liquid Chromatogr-amtandem massa spectrometry (UPLC/MS/MS)" have modified to "Ultra-performance liquid chromatography tandem mass spectrometry (UPLC/MS/MS)".

Point 2:Introduction line 47 - UPLC/MS/MS not HPLC/MS/MS

Response 2:     The “HPLC/MS/MS” has modified to” UPLC/MS/MS”.

Point 3:Figure 3 - explain the figure what refers, for example with letter and explain the meaning of each chromatogram.

Response 3:  Figure 3 has been modified.

Author Response

Point 1 :“super-high-performance liquid Chromatogr –amtandem mass spectrometry (UPLC / MS / MS)” should be replaced with “ultra-high performance liquid chromatographytandem mass spectrometry (UPLC-MS/MS)”

Response 1: The"Super-high-performance liquid Chromatogr-amtandem massa spectrometry (UPLC/MS/MS)" has modified to "Ultra-performance liquid chromatography tandem mass spectrometry (UPLC/MS/MS)".

Point 2It is difficult to understand what the authors want to say with “metabolic rule” maybe “metabolic route”

Response 2: The “metabolic rule” has modified to “metabolic route”

Point :3The nomenclature “parent” and “daughter” is not recommended since many years ago because the IUPAC tend to avoid the use of familiar relation. Then, the term recommended is “precursor” and “product”. Please, check the manuscript and replace it.

Response 3: Daughter ion has modified to product ion.

Round 2

Reviewer 1 Report

The manuscript has been relatively well revised as following reviewers comments, ans it is now suitable for the publication in Molecules, I think. 

-The end-

Author Response

Point :The title of the paper I misleading because it uses the term metabolism and metabolites/degradation products are not looked at.  Recommend changing to "Persistence of Emamectin Residues in Rice and the Environment"

Response The title has been modified to “Dissipation of Emamectin Benzoate Residues in Rice and Rice Environment”.

Point:Generally units of mg/kg are used for pesticide analysis, so use this throughout. There is a lot of inconsistence in this paper in this regard ideally should be mg kg-1.

Response The units has been modified.

Point:In the Abstract two concentration ranges are reported for calibration in solution and matrix.  These look odd, I guess that there should be a five-fold difference in the ranges.  Think that the solvent one should be 0.5 to 250. 

Response:Samples that exceed the linear range during the test must be diluted before analyzed by LC-MS/MS

Point: L25 suggest neurotoxic rather than nerve agent.  Nerve agent implies that it is used in chemical warfare.

ResponseThe”nerve agent” has been modified to“neurotoxic”.

Point:L35-36  Not sure what you are trying to say here.  Simply MRLs have been established or This active ingredient has not been licensed/authorised in many countries????

Response: The Cited literature shows the strong toxicity of Emamectin benzoate, which explains the need for MRLs in various countries.

Point:Can you also advise what the marker residues for emamectin rice is?

Does it include the 8,9-z-isomer?  If so, this limitation should be discussed in this work.  This isomer photodegradtion product that transforms back to the same peak as the parent isomer after the derivatisation used in the fluoresencence method but not in LC-MS/MS methods.  Therefore, the presence of a peak with similar ions may indicate photodegradration as the mass of the parent would be the same (double check versus literature for such papers). Also it would be good to show a chromatogram of a real sample also rather than just a standard.  

Response: The JMPR (2011) report and China's national standards stipulate that the residue marker of Methylavermectin benzoate is methaverin B1a. In this test, only B1a was concerned, and no photolysis products were involved. 8 , 9-z isomer (8,9-ZMa).The main peak in the map is abamectin B1a, which does not contain the 89-isomer of B1a. In the actual sample measured, the spectrum of the detected sample is similar to that of the standard. Only the main peak of B1a was found, but 8,9-ZMa was not found. The reason may be that the content of methylaminoavermectin B1a is low and the There are many degradation pathways of the sample, including photolysis, hydrolysis, microbial degradation, etc. The degradation products are complex, and the 8,9-ZMa content is too low to be detected.